# Electronic Structures of Monolayer Binary and Ternary 2D Materials: MoS_2_, WS_2_, Mo_1−*x*_Cr*_x_*S_2_, and W_1−*x*_Cr*_x_*S_2_ Using Density Functional Theory Calculations

**DOI:** 10.3390/nano13010068

**Published:** 2022-12-23

**Authors:** Chieh-Yang Chen, Yiming Li, Min-Hui Chuang

**Affiliations:** 1Parallel and Scientific Computing Laboratory, National Yang Ming Chiao Tung University, Hsinchu 300093, Taiwan; 2Institute of Communications Engineering, National Yang Ming Chiao Tung University, Hsinchu 300093, Taiwan; 3Institute of Biomedical Engineering, National Yang Ming Chiao Tung University, Hsinchu 300093, Taiwan; 4Department of Electronics and Electrical Engineering, National Yang Ming Chiao Tung University, Hsinchu 300093, Taiwan; 5Center for mmWave Smart Radar System and Technologies, National Yang Ming Chiao Tung University, Hsinchu 300093, Taiwan

**Keywords:** two-dimensional materials, molybdenum disulfide, tungsten disulfide, density functional theory, density of state, band structure, binary 2D materials, ternary 2D materials, chromium, mole fraction

## Abstract

Two-dimensional (2D) materials with binary compounds, such as transition-metal chalcogenides, have emerged as complementary materials due to their tunable band gap and modulated electrical properties via the layer number. Ternary 2D materials are promising in nanoelectronics and optoelectronics. According to the calculation of density functional theory, in this work, we study the electronic structures of ternary 2D materials: monolayer Mo_1−*x*_Cr*_x_*S_2_ and W_1−*x*_Cr*_x_*S_2_. They are mainly based on monolayer molybdenum disulfide and tungsten disulfide and have tunable direct band gaps and work functions via the different mole fractions of chromium (Cr). Meanwhile, the Cr atoms deform the monolayer structures and increase their thicknesses. Induced by different mole fractions of Cr material, energy band diagrams, the projected density of states, and charge transfers are further discussed.

## 1. Introduction

Transition-metal dichalcogenide (TMD) in the two-dimensional material family has received much attention owing to the direct band gap of its monolayer structure and great process stability [1,2,3,4,5,6,7]. Different band gaps of monolayer MoS_2_ have been reported, such as 2.12 eV using the HSE06 method [8], 1.88 eV using the GVJ-2e method [9], and 1.90 eV from an experiment [10]. For the band gaps of monolayer WS_2_, 2.03 eV using the GVJ-2e method [9] and 2.01 eV from an experimentally measured method [11] were determined. Various approaches have been reported to modify the properties of TMD materials, such as doping techniques [12,13,14,15,16], multiple TMD alloy materials [17,18,19], etc. Nanomaterials with relatively high process stability, simple material composition, and a tunable band gap are critical in the fields of semiconductor electronics and optical devices. To meet the aforementioned requirements, in addition to binary 2D materials, novel ternary 2D materials have recently been of great interest [17,18,19,20,21]. Regarding the ternary TMD materials, the early works of Xi et al. [22] and Chen et al. [23] have investigated the Mo_1−*x*_W*_x_*S_2_ alloy, where *x* is the mole fraction. Furthermore, the Mo_1−*x*_W*_x_*S_2_ and MoS_2(1−*x*)_Se_2*x*_ alloys, both with excellent uniformity and controllable compositions, were synthesized recently [19]. These reported works have revealed the potential of exchanging different transition metals and the interchangeability of sulfur and selenium for certain TMD materials. Although different applications of MoS_2_-based ternary materials for photocatalytic and battery have been experimentally studied [17,18], the electronic structures of ternary TMD materials with different compositions of transition metals have not been well studied yet. Therefore, by using density functional theory (DFT), we computationally estimated the energy band diagrams and projected the density of states and charge transfers of monolayer (ML-) Mo_1−_*_x_*Cr*_x_*S_2_ and ML-W_1−_*_x_*Cr*_x_*S_2_ under different mole fractions, where Mo, W, Cr, S, and *x* are molybdenum, tungsten, chromium, sulfur, and the mole fraction of Cr material, respectively. 

This paper is organized as follows. Section 2 elaborates on the simulation flow and settings of DFT. Section 3 briefly discusses the simulation results. Finally, we conclude this study and suggest future works.

## 2. Materials and Methods

Figure 1a,b,a′,b′ shows the structures of ML-Mo_1−*x*_Cr*_x_*S_2_ and ML-W_1−*x*_Cr*_x_*S_2_ with the mole fraction, *x*, equal to 0.25 and 0.0625, respectively. The purple balls represent Mo or W atoms, the yellow balls represent S atoms, and the blue balls represent Cr atoms. The areas enclosed by black straight lines in Figure 1a,a′ are the simulated cells, which were built from ML-MoS_2_ or ML-WS_2_ supercells, 2 × 2 (*x* = 0.25) and 4 × 4 (*x* = 0.0625) in size, respectively, with one Mo or W atom replaced by one Cr atom. The simulations of the ML-MoS_2_ and ML-WS_2_ were firstly achieved and examined to assure the reasonable relaxations of the ternary 2D materials [24,25]. To decouple the periodic images between adjacent monolayers, a 15 Å-thick vacuum layer was applied along the C-axis for all of the studied materials. The structural relaxation and electronic structure were calculated using the Vienna ab initio Simulation Package (VASP) [26] under spin-polarized DFT. The Perdew–Burke–Ernzerhof (PBE) method [27] was used as the exchange-correlation functional and the cutoff kinetic energy was 550–600 eV. The accuracy of simulations for ML-MoS_2_ and ML-WS_2_ was examined in our earlier works [24,25]. As listed in Figure 1c, the Brillouin zone (BZ) was sampled with 18 × 18 × 1, 4 × 4 × 1, and 10 × 10 × 1 Γ-centered *k*-meshes under the Monkhorst–Pack scheme [28] for ML-Mo_1−*x*_Cr*_x_*S_2_ and ML-W_1−*x*_Cr*_x_*S_2_, with *x* = 0, 0.0625 and 0.25, respectively. For all investigated materials, the tolerance of force acting on each atom for the structure relaxation was smaller than 5 × 10^−4^ eV/Å; the convergence criterion for the self-consistent electron energy was set as 10^−8^ eV.

## 3. Results

The optimized values of the lattice parameters and the formation energy for the studied materials are listed in Table 1. The distance, d_M−S_, is from the Mo or W atom to their neighbor S atoms; θ is the angle between the Mo or W and the adjacent S atoms above and below it. For the monolayer Mo_1−*x*_Cr*_x_*S_2_ and W_1−*x*_Cr*_x_*S_2_, where *x* is not zero, these parameters referred to the Cr atom and its adjacent S atoms. Notably, the results indicate that a higher mole fraction of the Cr induced a larger monolayer thickness. The bond lengths of Cr and S were shortened, and the θ of the Cr atom and the adjacent S atoms increased, so the local thickness was reduced. Consequently, the largest monolayer thickness occurred at the position far from the Cr atom due to the balance of stress in the material. The formation energy of a unit cell is given by
(1)$$E_{form}=x(E_{\mathrm{Mo}(1-x)\mathrm{Cr}(x)S2}-E_{\mathrm{MoS}2}+E_{\mathrm{Mo}}-E_{\mathrm{Cr}}),$$
where *E*_Mo(1−*x*)Cr(x)S2_ and *E*_MoS2_ are the total energies of the doped and pristine materials for a 2 × 2 or 4 × 4 ML-Mo_1−*x*_Cr*_x_*S_2_ supercell, and both *E*_Mo_ and *E*_Cr_ are the total energies of the Mo and Cr atoms. W_1−*x*_Cr*_x_*S_2_ had a similar expression. Notably, the calculated *E_form_* of the unit cell of ML-Mo_0.9375_Cr_0.0625_S_2_ was 0.19 eV which is close to the estimated value of 0.21 eV (the formulation energy of 4 × 4 = 16 cells was 3.32 eV/16 = 0.21 eV) in [29]. As listed in Table 1, the formation energies of the explored materials indicate that it was difficult to form the material with a high doping concentration of Cr atoms. Furthermore, compared to ML-W_1−*x*_Cr*_x_*S_2_, under the same mole fraction, ML-Mo_1−*x*_Cr*_x_*S_2_ had a lower *E_form_*, so it was more easily formed. To examine the stability of the material, we further estimated the formation energy to form the material with vacancies from the doped material. The formation energies of a unit cell were calculated using the following formula:(2)$$E_{form,doped\to Vacancy}=x(E_{\mathrm{Mo}(1-x)\mathrm{Vacancy}(x)S2}-E_{\mathrm{Mo}(1-x)\mathrm{Cr}(x)S2}+E_{\mathrm{Cr}})$$

The formation energies were 2.50 and 0.65 eV for ML-Mo_1−*x*_Cr*_x_*S_2_ with *x* = 0.25 and 0.0625, respectively. Similarly, for ML-W_1−*x*_Cr*_x_*S_2_ with *x* = 0.25 and 0.0625, they were 2.43 and 0.63 eV, respectively. Notably, all calculations were positive. This implies that the Cr tended to stay in the materials. The electronic band structures of the monolayer Mo_1−*x*_Cr*_x_*S_2_ and W_1−*x*_Cr*_x_*S_2_ with *x* = 0, 0.0625 and 0.25, respectively, are shown in Figure 2a,a″,b,b″. The valence band maximum was set to zero as the reference energy level. All of the bands are indicated by black lines, and the projected bands contributed by the Cr are marked by purple circles, where the circle size indicates the weight of the contribution. All studied materials, ML-MoS_2_, ML-WS_2_, ML-Mo_1−*x*_Cr*_x_*S_2_, and ML-W_1−*x*_Cr*_x_*S_2_, show direct band gaps at the K point of the BZ. Their band gap values are noted in Figure 2a,a″,b,b″. As the results of the band gap values are in the range of 1.28–1.84 eV, the ML-Mo_1−*x*_Cr*_x_*S_2_ and ML-W_1−*x*_Cr*_x_*S_2_ were semiconductors. We studied the energy bands of ML-Mo_1−*x*_Cr*_x_*S_2_ and ML-W_1−*x*_Cr*_x_*S_2_; for materials that were different from the monolayer, such as bilayers and bulk, their band gaps were indirect and reduced [15]. The projected densities of states (PDOS) of the ML-Mo_1−*x*_Cr*_x_*S_2_ and ML-W_1−*x*_Cr*_x_*S_2_ for different mole fractions of Cr are shown in Figure 3a,a″,b,b″. Figure 3a,b shows the binary ML-MoS_2_ and ML-WS_2_. Their Mo and W atoms offered major PDOS near the band edges. On the other hand, as presented in Figure 3a′,a″,b′,b″, the Cr atoms contributed additional energy states near the edges of the valence and conduction bands. In particular, the PDOS resulting from the Cr at the edge of the conduction band was more than what resulted from the Mo or W atoms. Thus, the widths of the band gaps were dominated by Cr atoms.

According to the calculated work function of the studied materials, Figure 4a,b shows the energy levels of the conduction band minimum (CBM) and the valence band maximum (VBM). The CBM energy level of the ML-WS_2_ was higher than that of the ML-MoS_2_ due to its relatively higher band gap. For different mole fractions of the Cr, both the ML-Mo_1−*x*_Cr*_x_*S_2_ and ML-W_1−*x*_Cr*_x_*S_2_ had lower CBM energy levels for larger mole fractions; meanwhile, their VBM energy levels were marginally altered. Compared to the ML-Mo_1−*x*_Cr*_x_*S_2_, as shown in Figure 4a, the changes in the energy levels of the CBM and VBM for the ML-W_1−*x*_Cr*_x_*S_2_ were significant, as shown in Figure 4b. Furthermore, Figure 5a–c illustrates the values of the direct band gaps, work functions, and charge transfers of the Cr atom with respect to different mole fractions, respectively. Figure 5a plots the trend of the direct band gaps versus the mole fractions. To capture the sharp variation in the band gap, an additional mole fraction of 0.11 was further simulated, as shown in Figure 5a. The reduction of the band gap with respect to the increase in the mole fraction of the Cr was nonlinear, and the effect was saturated when the mole fraction of the Cr was around *x* = 0.15. Figure 5b indicates that the work function and the mole fraction of the Cr were positively correlated. As shown in Figure 5c, the charge transfers of the Cr Δq_Cr_ were obtained via the Bader charge analysis [30]. Then, they were normalized by the number of the MoS_2_ (or WS_2_) in the ML-Mo_1−*x*_Cr*_x_*S_2_ (or ML-W_1−*x*_Cr*_x_*S_2_). The negative values show the loss in the electron charge. The results indicate that the charge loss increased as the mole fraction increased. Thus, to further study the charge distribution in the materials, as shown in Figure 6, we calculated the charge difference, CHG_diff_, by the formula given below: (3)$${\mathrm{CHG}}_{\mathrm{diff}}={\mathrm{CHG}}_{\mathrm{ML}-\mathrm{Mo}(1-x)\mathrm{Cr}(x)S2}-{\mathrm{CHG}}_{\mathrm{ML}-\mathrm{MoS}2}-{\mathrm{CHG}}_{\mathrm{Cr}},$$
where CHG_ML-Mo(1−*x*)Cr(*x*)S2_, CHG_ML-MoS2_, and CHG_Cr_ are the charge distributions of the simulated ML-Mo_1−*x*_Cr*_x_*S_2_, ML-MoS_2_ (i.e., the ML-Mo_1−*x*_Cr*_x_*S_2_ without considering the Cr atom), and the Cr atom, respectively. Similarly, these calculations were carried out for the ML-W_1−*x*_Cr*_x_*S_2_. The green and yellow colors represent the loss and gain of an electron charge, respectively. The cutting planes along the a- and b-directions (see Figure 5) at the height of the Cr atom are presented. The Cr atom contributed additional electron charges to its surrounding atoms. Notably, the Mo atoms near the Cr atom received more electron charges for a relatively higher mole fraction of the Cr.

## 4. Conclusions

In this work, the geometry and electronic structures of ML-MoS_2_, ML-WS_2_, ML-Mo_1−*x*_Cr*_x_*S_2_, and ML-W_1−*x*_Cr*_x_*S_2_ were discussed. Their band structures imply that they behave as semiconductors. Notably, the variations in the direct band gap and the work function of the explored ML-Mo_1−*x*_Cr*_x_*S_2_ and ML-W_1−*x*_Cr*_x_*S_2_ with different mole fractions were nonlinear. These characteristics could be adjusted by tuning the mole fraction; however, the effect was saturated when *x* was greater than 0.15, where the energy levels of the CBM and VBM with respect to *x* were summarized. On the other hand, to clarify how the Cr atoms in the ML-Mo_1−*x*_Cr*_x_*S_2_ and ML-W_1−*x*_Cr*_x_*S_2_ interacted with the surrounding atoms, the electron-charge differences were estimated and discussed. The results of this study benefit the design and optimization of monolayer ML-Mo_1−*x*_Cr*_x_*S_2_ and ML-W_1−*x*_Cr*_x_*S_2_ for future optoelectronics and nanoelectronics. To consider more accurate correlation effects among the atoms, the DFT scheme, including further adjustments and the hybrid functionals could be advanced in our future work. 

## Figures and Tables

**Figure 1 nanomaterials-13-00068-f001:**
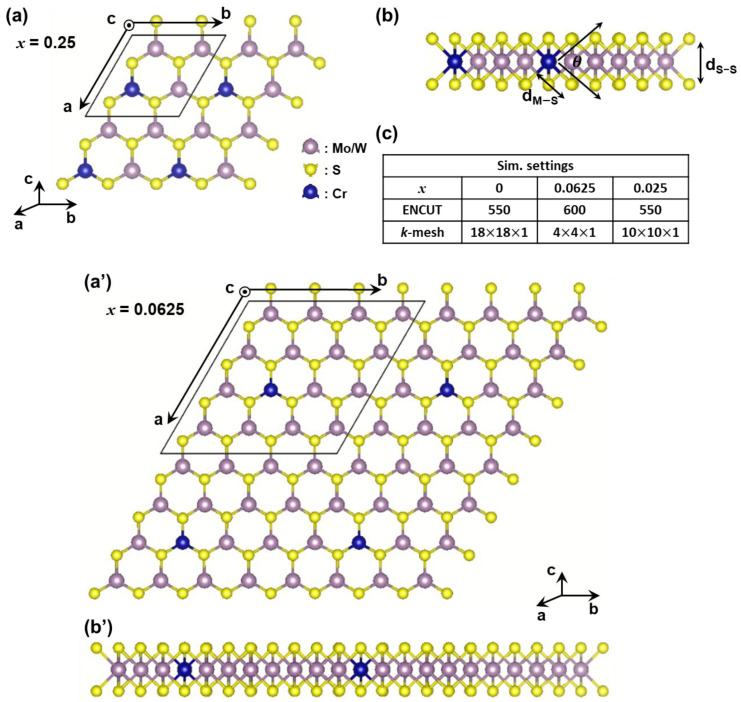
Top view and side view of monolayer M_1−*x*_Cr*_x_*S_2_ (M = Mo, W) with different compositions: (**a**,**b**) *x* = 0. 25 and (**a**′,**b**′) *x* = 0.0625. For *x* = 0. 25 and 0.0625, one of the Mo or W atoms was replaced by one Cr atom in a MoS_2_ or WS_2_ supercell monolayer, with sizes of 2 × 2 and 4 × 4, respectively. Notably, d_S−S_ is the material thickness. (**c**) The critical simulation settings for different mole fractions.

**Figure 2 nanomaterials-13-00068-f002:**
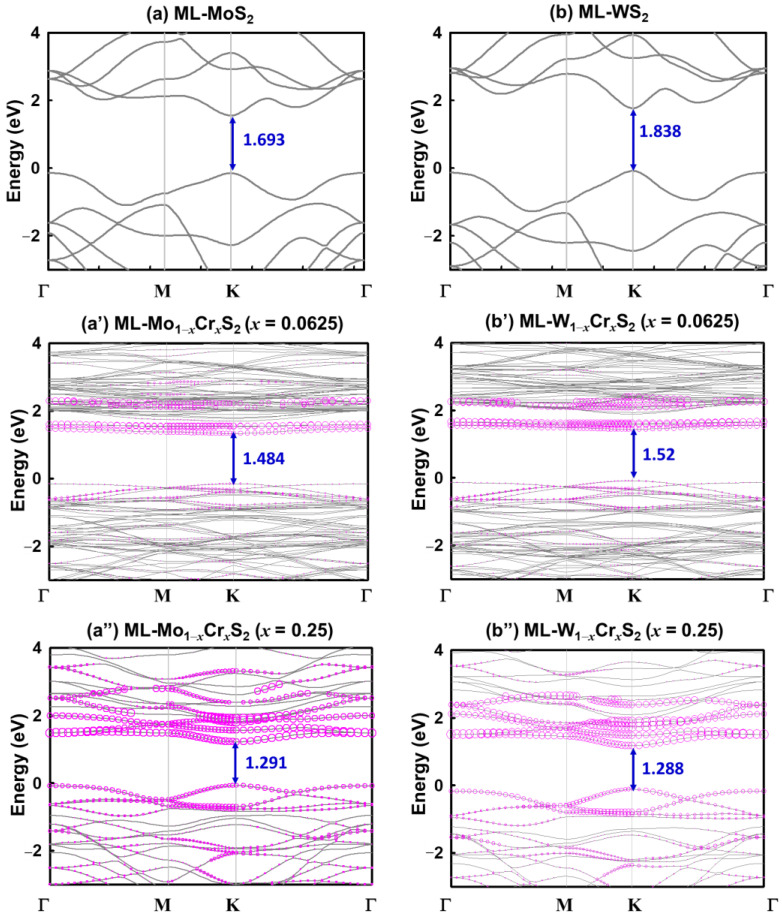
The simulated energy bands of (**a**) ML-MoS_2_ and (**b**) ML-WS_2_. (**a**′,**a**″) ML-Mo_1−*x*_Cr*_x_*S_2_ and (**b**′,**b**″) W_1−*x*_Cr*_x_*S_2_ materials with respect to different mole fractions. The zero energy was set to the valence band maximum. The black lines and purple circles are all bands and projected bands contributed by the Cr element, respectively. The circle sizes of the projected bands indicate the relative weighting. All of the band structures of ML-Mo_1−*x*_Cr*_x_*S_2_ and ML-W_1−*x*_Cr*_x_*S_2_ are direct band gap semiconductors.

**Figure 3 nanomaterials-13-00068-f003:**
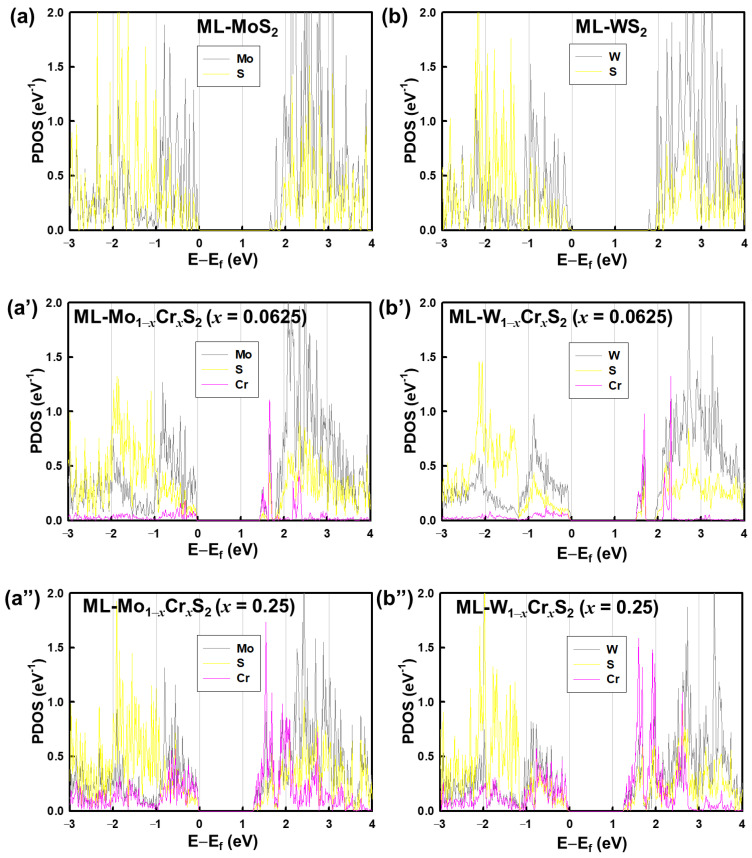
The projected densities of states (PDOS) of (**a**,**a**′,**a**″) ML-Mo_1-*x*_Cr*_x_*S_2_ and (**b**,**b**′,**b**″) ML-W_1-*x*_Cr*_x_*S_2_ with respect to each element and different mole fractions, where the valence band maximum was set to zero. The values of PDOS were normalized by the supercell size, where the sizes were 16 and 4 for *x* = 0.0625 and 0.25, respectively. The PDOS contributed by the Cr are mainly near the bottom of the conduction bands, so the band gaps of the ML-Mo_1−*x*_Cr*_x_*S_2_ and ML-W_1−*x*_Cr*_x_*S_2_ were reduced.

**Figure 4 nanomaterials-13-00068-f004:**
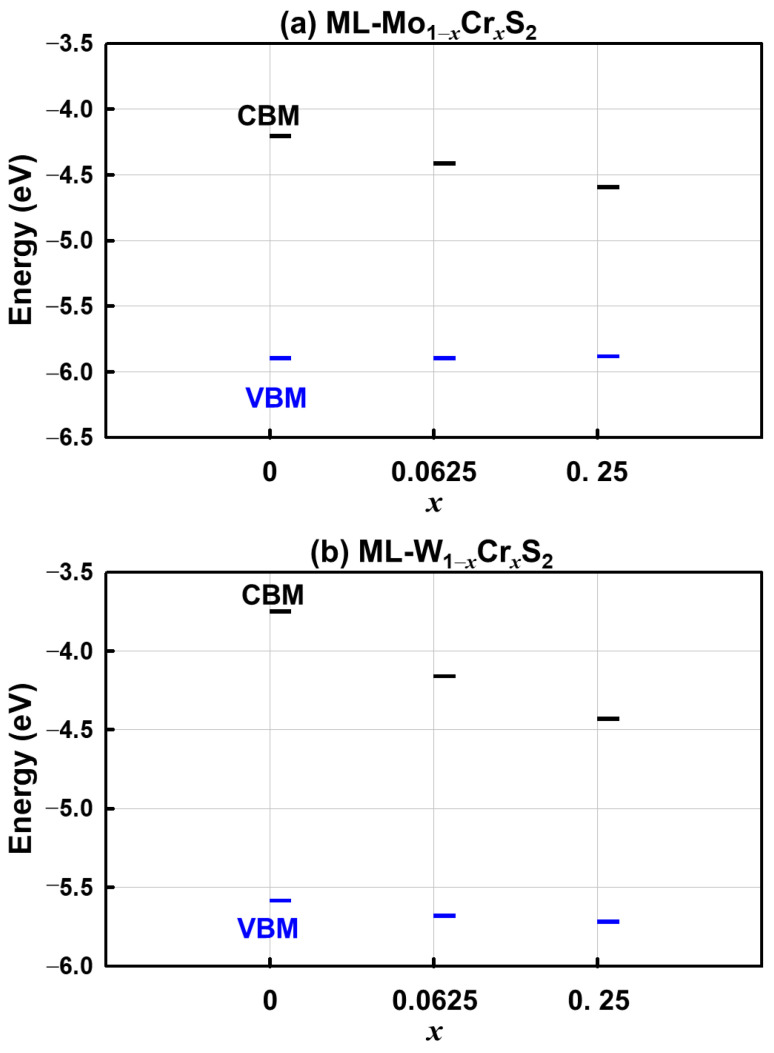
The energy levels of CBM and VBM for the studied materials: (**a**) ML-Mo_1−_*_x_*Cr*_x_*S_2_ and (**b**) ML-W_1−_*_x_*Cr*_x_*S_2_. The levels of CBM decreased when the mole fractions increased; meanwhile, the level changes of the VBM were relatively small. Because of the large Cr contribution near the CBM, the gap shift occurred mostly in the CB rather than the VB.

**Figure 5 nanomaterials-13-00068-f005:**
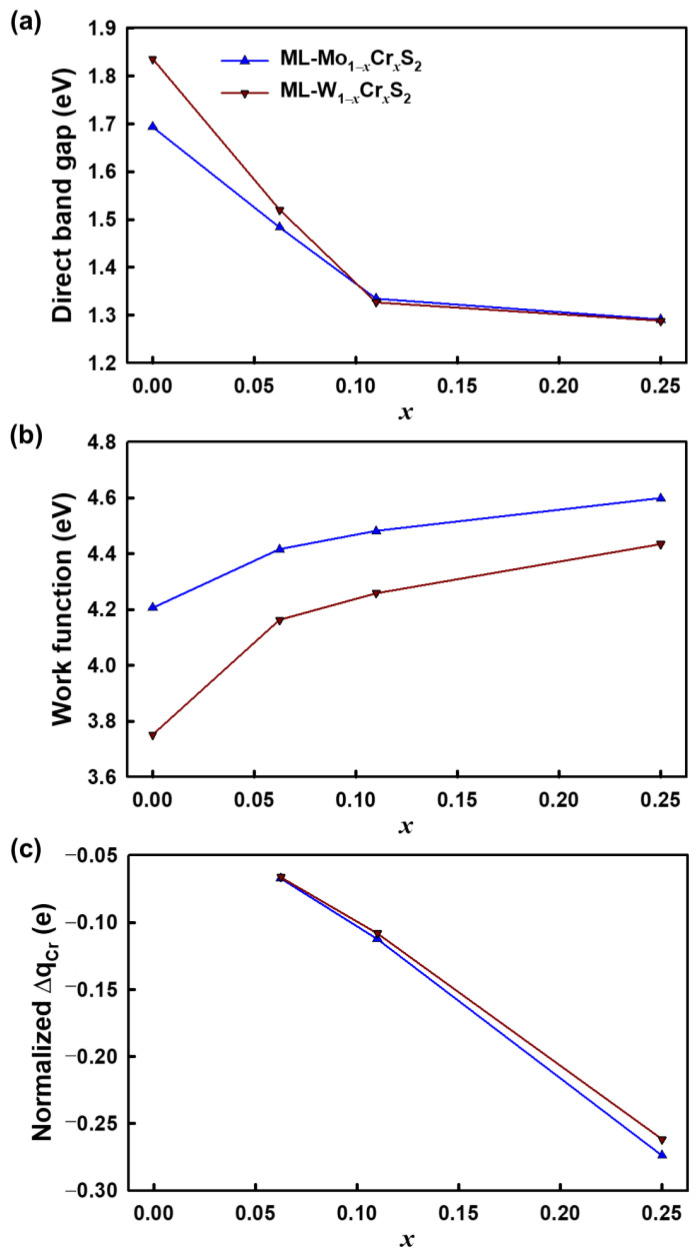
The values of (**a**) direct band gaps, (**b**) work functions, and (**c**) charge transfers of the Cr for the studied materials with respect to different mole fractions. (**a**) Additional simulations were considered for the mole fraction of 0.11 to provide the best estimation of the energy band gap. The results indicate that the variation with respect to the mole fraction of Cr was nonlinear, and the effect was marginal for a large mole fraction. (**b**) The work function increased when the composition of the Cr increased. (**c**) The electron-charge transfer of the Cr atom was analyzed via Bader charge analysis [30]; negative values indicate the loss of an electron charge. The unit e is the elementary charge of an electron. The variations in the charge transfer were negligible for different mole fractions.

**Figure 6 nanomaterials-13-00068-f006:**
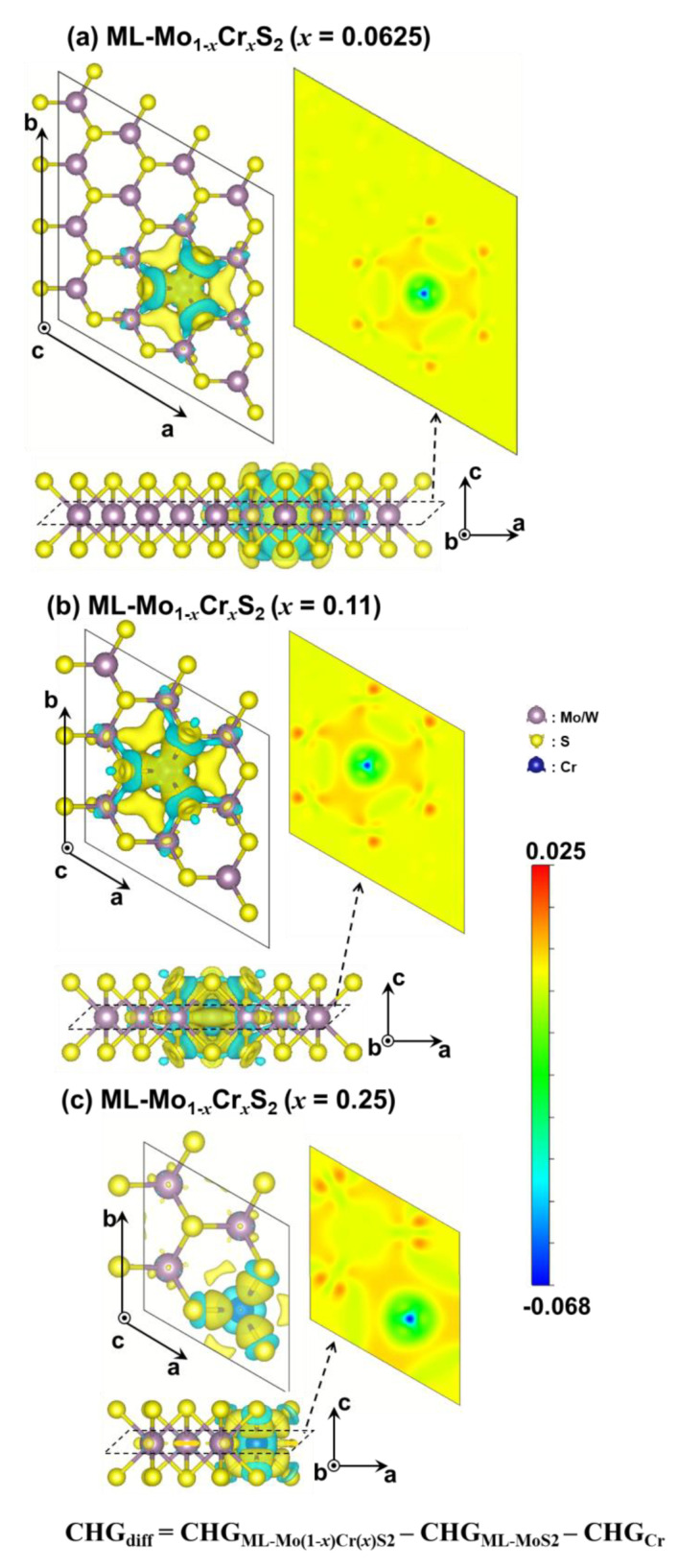
The charge transfer induced by the Cr atom in the ML-Mo_1-*x*_Cr*_x_*S_2_ under low and high mole fractions: (**a**) *x* = 0.0625, (**b**) *x* = 0.11, and (**c**) *x* = 0.25. The distributions of the charge difference were calculated by the following formula: CHG_diff_ = CHG_ML-Mo(1−*x*)Cr(*x*)S2_ – CHG_ML-MoS2_ – CHG_Cr_. The isosurfaces of the green and yellow colors indicate the loss and gain of electron charges, respectively. The plots of the cutting planes are at the height of the Cr atoms; the color bar represents the cutting planes. The distributions show that the S and Mo atoms near the Cr atom received extra electron charges owing to the repulsed electron charge by the Cr atom.

**Table 1 nanomaterials-13-00068-t001:** List of the final optimized lattice parameters and the formation energy for ML-Mo_1−*x*_Cr*_x_*S_2_ and ML-W_1−*x*_Cr*_x_*S_2_ with respect to different mole fractions. For notations in the first column, the subscript M is Mo, W, and Cr atoms of ML-MoS_2_, ML-WS_2_, and their composites, including the Cr material, respectively. The normalized a is the a divided into the number of MoS_2_/WS_2_ along the a-direction.

Explored Material	ML-Mo_1−*x*_Cr*_x_*S_2_			ML-W_1−*x*_Cr*_x_*S_2_		
Mole Fraction *x*	0	0.0625	0.25	0	0.0625	0.25
a (Å)	3.18	12.697	6.299	3.181	12.705	6.307
Normalized a (Å)	3.18	3.174	3.149	3.181	3.176	3.154
d_M−S_ (Å)	2.413	2.325	2.329	2.419	2.334	2.338
Thickness (Å)	3.131	3.143	3.169	3.148	3.152	3.169
θ (deg)	80.9	81.683	81.983	81.185	81.7	82.086
*E_form_* (eV)		0.19	0.75		0.31	1.22

## Data Availability

Not applicable.

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
