# Peer review of "Electronic Structures of Monolayer Binary and Ternary 2D Materials: MoS2, WS2, Mo1−xCrxS2, and W1−xCrxS2 Using Density Functional Theory Calculations"

_nanomaterials, 2022, doi:10.3390/nano13010068_

Round 1

Reviewer 1 Report

The authors use density functional theory (DFT) calculations to study the electronic structure (in particular band gaps) of Cr doped MoS2 and WS2 monolayers. They observe a reduction of the gap as function of Cr concentration, saturating at about 15%, and a slight increase of the work function. Some minor structural changes are observed upon doping.

There are already several studies of doped transition metal dichalcogenides with the aim to tailor the band gap, however, not so many on Cr doping. Unfortunately, there are several technical concerns that make me hesitant to believe in the shown results:

From the densities of states (DOS) displayed in fig.3 I conclude that Cr is non-magnetic here. In the literature [RSC Advances 7, 20116 (2017)] I read that, although the Cr doped MoS2 monolayer has a total magnetic moment of zero, the Cr atoms show a magnetization of 2.6 Bohr magneton (this is compensated by surrounding atoms). This is different from the present calculation and affects also the states around the band gap.  

Maybe this comes from the fact that the authors did not account for correlation effects in the dopant, usually for 3d atoms these cannot be neglected. A simple DFT+U scheme would already help to improve the description. Alternatively (and this would probably also improve the band gaps) hybrid functionals could be used.

Another issue concerns the band structures (fig.2): In the calculation of supercells, naturally a backfolding of the Brillouin zone (BZ) appears. Experimentally, the selection of transitions is more naturally described in the unfolded BZ (i.e. that of the primitive 1x1 unit cell). Therefore, it would be better to show the unfolded band structures.

To estimate the stability of the doped layers, the authors should calculate the defect formation energies. In principle, they should have all data available from the calculations of the charge transfer (fig.6).

In the band structure and DOS of the low-doped layers (x=1/16), I did not see the occupied d-states of Cr. The energy range of these plots should be enlarged to show them.

In fig.5a the authors show data from an x=1/9 calculations, it would be good to show that also in the other panels. Also in panel (c) the charge transfer from Mo(W) for x=0 could be included for comparison.

Abstract, line 17: "moderate the electrical properties" - maybe "modulate" was meant?
Conclusions, line 175: "x is greater that 0.15" - maybe "smaller"?

Author Response

See the attached response letter.

Reviewer 2 Report

The paper presents standard DFT calculations for a few ternary 
transition metal dichalcogenides, Co1-xCrxS2 and W1-xCrxS2 with  x=0.0625 and 0.25. Cr is in the same column in the periodic table as W and Mo and thus isovalent so this makes sense. The main finding is a decrease in band gap by adding Cr.  The paper is publishable with a few minor corrections.  

1) Since all these materials are non-magnetic, why do the authors show both spin up and spin down PDOS? one might as well only show the total PDOS which would simplify Fig. 3. 
2) In terms of the charge transfer the authors mention that it is opposite in 
Mo and W case but they provide a figure only for Mo and say it is negligibe; in both cases. Why is it considered negligible? This seems a rather arbitrary judgement.  On what is it based?  It would be more relevant to comment on the nature of the charge transfer. From their definition it seems to look for charge transfer between Cr and the Mo or W. Is the charge density of Cr which  is used here calculated for a separate free atom? or is it with respect to some The Bader basin defined around the Cr? This part is not very clearly worded.  The relation of the charge transfer shown in Fig. 6 with respect t the d-orbitals should be explained more clearly. One would perhaps expect the 3d orbitals of Cr to lie deeper than the 4d or 5d of Mo and W. Is this the reason for a net charge transfer toward Cr? 

From the band figures, it seems there is a larger Cr contribution near the CBM than in the VBM. Is this the reason why the gap shift occurs mostly in the CB rather than the VB?

Can the authors add a few sentences about the expected accuracy of DFT for these materials? Gaps are usually underestimated. Perhaps DFT is sufficient to explore the difference in gap due to Cr but what is the state of the literature of MoS2 with respect to band gaps calculated at higher levels such as GW approximation or hybrid functionals? 

In table 1 instead of giving the a lattice constant of the 2x2 or 4x4 cell, it would be better to give the a/2 or a/4 so as to directly compare with the 1x1 cell . 

The authors find that near the Cr the bond length is reduced but the angle is increased. It is still surprising to me that the overall thickness 

as measured by d_S-S is increasing when adding Cr.  Perhaps it is worthwhile showing this distance as function of lateral position, so as fucntion of distance to the Cr?  

A few typos. page 2 line 69 I suppose the authors want to say Gamma-centered k-mesh but instead of Gamma there seems to be some other strange character. 

p. 1 line 31 has drawn many attentions would better sound as has received much attention 

Author Response

Please see the attached response letter

Reviewer 3 Report

This work presents the “Electronic Structures of Monolayer Binary and Ternary 2D Materials: MoS2, WS2, Mo1-xCrxS2, and W1-xCrxS2 Using Density 3 Functional Theory Calculations”. Authors studied electronic structures of ternary 2D materials: monolayer Mo1-xCrxS2 and W1-xCrxS2. This manuscript is well written and presented appropriately. Therefore, I recommend for the publication in nanomaterials after correcting grammatical errors.

Author Response

Please see the attached response letter

Round 2

Reviewer 1 Report

The authors replied to my concerns, partially this solves the problems. Nevertheless, there are a few points to address (I refer here to the numbering of the authors):

Ad 1: If the authors obtained non-magnetic solutions they should write that and then in fig.3 it is sufficient to show just one spin-channel.

Ad 2: Line 200 "correlation effects among the atoms": Do they really mean "among" (intersite terms needed) or "at" (on-site terms sufficient)?

Ad 4: Please notice that the formation energies are orders of magnitude too high.  Maybe this is because the formula should be
E_form = E_Mo(1-x)Cr(x)S2 - E_MoS2 + x(E_Mo - E_Cr)
likewise, E_Cr should be x E_Cr in eq.2. Compare these energies also to Physica Scripa 97, 095805 (2022) where they are in the order of 3.3 eV.
In particular this point has to be resolved.

Another reference that should be compared to is J. Supercond. Nov. Mag. 34, 3441 (2021). This is cited as [15] but no comparison for the band structure / gaps of Cr doped MoS2 is made.
